# Proton Pump Inhibitors and Likelihood of Colorectal Cancer in the Korean Population: Insights from a Nested Case–Control Study Using National Health Insurance Data

**DOI:** 10.3390/cancers15235606

**Published:** 2023-11-27

**Authors:** Mi Jung Kwon, Kyeong Min Han, Joo-Hee Kim, Ji Hee Kim, Min-Jeong Kim, Nan Young Kim, Hyo Geun Choi, Ho Suk Kang

**Affiliations:** 1Department of Pathology, Hallym University Sacred Heart Hospital, Hallym University College of Medicine, Anyang 14068, Republic of Korea; mulank99@hallym.or.kr; 2Hallym Data Science Laboratory, Hallym University College of Medicine, Anyang 14068, Republic of Korea; 3Division of Pulmonary, Allergy, and Critical Care Medicine, Department of Medicine, Hallym University Sacred Heart Hospital, Hallym University College of Medicine, Anyang 14068, Republic of Korea; luxjhee@hallym.or.kr; 4Department of Neurosurgery, Hallym University Sacred Heart Hospital, Hallym University College of Medicine, Anyang 14068, Republic of Korea; kimjihee.ns@hallym.or.kr; 5Department of Radiology, Hallym University Sacred Heart Hospital, Hallym University College of Medicine, Anyang 14068, Republic of Korea; drkmj@hallym.or.kr; 6Hallym Institute of Translational Genomics and Bioinformatics, Hallym University Medical Center, Anyang 14068, Republic of Korea; 7Suseo Seoul E.N.T. Clinic and MD Analytics, Seoul 06349, Republic of Korea; mdanalytics@naver.com; 8Department of Internal Medicine, Hallym University Sacred Heart Hospital, Hallym University College of Medicine, Anyang 14068, Republic of Korea

**Keywords:** colorectal cancer, proton pump inhibitor, national healthcare data, nested case–control study

## Abstract

**Simple Summary:**

We conducted a study using the Korean National Health Insurance Service–National Sample Cohort database to investigate if a hazard connection existed between prior use of proton pump inhibitors (PPIs) and the development of colorectal cancer (CRC) in the Korean population. Our research, which employed a nested case–control study with a statistical technique known as “propensity score nested weighted multivariate logistic regression”, revealed that the risk of developing CRC increased regardless of whether individuals had a history of PPI use or the duration of PPI use. This suggests that prior use of PPIs, whether currently or in the past, and regardless of the duration, may be linked to an increased likelihood of CRC within the Korean population, highlighting the need for cautious and precise adherence to PPI medications according to treatment guidelines to mitigate potential adverse effects.

**Abstract:**

The potential connection between proton pump inhibitors (PPIs) and colorectal cancer (CRC) risk remains unclear, with specific ethnic genetic backgrounds playing a role in PPI-induced adverse effects. In this nested case–control study, we investigated the risk of CRC in relation to preceding PPI use and the duration of use using data from the Korean National Health Insurance Service–National Sample Cohort database, including 9374 incident CRC patients and 37,496 controls. To assess the impact of preceding PPI exposure (past vs. current) and use duration (days: <30, 30–90, and ≥90) on incident CRC, we conducted propensity score overlap-weighted multivariate logistic regression analyses, adjusted for confounding factors. Our findings revealed that past and current PPI users had an increased likelihood of developing CRC. Regardless of duration, individuals who used PPIs also had higher odds of developing CRC. Subgroup analyses revealed that CRC occurrence increased independent of history or duration of prior PPI use, consistent across various factors such as age, sex, income level, and residential area. These findings suggest that PPI use, regardless of past or present use and duration of use, may be related to an increased risk of developing CRC in the Korean population.

## 1. Introduction

Colorectal cancer (CRC) is the third most frequently diagnosed cancer worldwide and is the second leading cause of cancer-related deaths, accounting for 10% of all cancer cases and fatalities [1]. In South Korea, the epidemiology of CRC has been undergoing significant transformations during recent decades, with the nation witnessing the world’s second-highest incidence rate of CRC in 2018, at 44.5 cases per 100,000 individuals annually [2]. This surge has placed a substantial burden on public health, making CRC the third most prevalent cancer in the country [2]. Notably, the prevalence of young-onset CRC in Korea has also meaningfully increased over the last two decades [3]. Several factors are believed to contribute to the increasing incidence and mortality trends of CRC in Korea, including sedentary lifestyles, reduced physical activity, diabetes mellitus, obesity, consumption of spicy foods, alcohol, red and processed meat, and cigarette smoking [4,5]. The recent surge in CRC incidence among Asians, including Koreans, has been closely tied to modifiable environmental factors [3,6,7,8,9]. Therefore, identifying potential risk factors for CRC is a fundamental approach to prevention. 

Proton pump inhibitors (PPIs) are medications globally utilized for their ability to effectively reduce gastric acid secretion [10]. Although PPIs are not over-the-counter drugs in South Korea, their usage has increased more than 1.5 times over the past five years [11]. They are commonly prescribed for acid-related conditions such as peptic ulcers, gastroesophageal reflux disease (GERD), and functional dyspepsia [10]. Histamine-2 receptor antagonists (H2-blockers), an alternative class of acid suppressant drugs, are also indicated for similar conditions, although they are less effective at lowering gastric acid levels than PPIs [12]. Despite their widespread use and general safety profile [12,13], concerns have emerged regarding potential adverse effects of PPIs, including the development of neuroendocrine tumors in rat oxyntic mucosa attributed to PPI-induced hypergastrinemia [14]. The discovery that ranitidine, an H2-blocker, is contaminated with N-nitroso dimethylamine, a potential carcinogen, has raised concerns about its potential to cause gastrointestinal cancer, including CRC [15]. However, the precise impact of concurrent use of PPIs and H2-blockers remains unclear [15,16,17,18], given that PPI users are also more likely to use H2-blockers [12]. 

There is a potential link between PPI use, hypergastrinemia, and an increased risk of gastrointestinal malignancies [19], including CRC [20]. Experimental research has shown that colon mucosa responds to the trophic effects of gastrin [21], leading to the progression of colon adenomas [22]. CRC often exhibits overexpression of the gastrin receptor with a 10-fold increase in gastrin-binding capacity compared to that of normal colonic epithelium [23]. Furthermore, PPI use can significantly impact the gut microbiome [24,25], potentially contributing to the development of gastrointestinal tumors [26]. A recent study revealed that PPI usage promotes the growth and metastasis of CRC by elevating gastrin levels and increasing yes-associated protein expression, ultimately leading to alterations in gut flora and a shift towards fecal alkalization [27].

Despite these plausible mechanisms, 11 epidemiological publications [16,17,18,28,29,30,31,32,33,34,35] and their associated meta-analyses [36,37,38] investigating the correlation between PPI use and CRC have generated contradictory findings. While some studies have suggested a potential connection between PPI usage and CRC, including in subgroup analyses regarding usage duration and prior history of use [18,29,31,32,34,37,39], others have reported no such association [28,30,38]. Notably, the findings from three prospective cohort studies even indicated a possible protective effect of PPI use against CRC [17]. Common issues in these studies were sample size imbalances and variations in demographic data (age, sex, socioeconomic status, and comorbidities) [16,17,28,29,30,31,32,34], introducing potential selection bias due to disease prevalence, urban–rural differences, and socioeconomic variations. A population-based study in Korea investigated the relationship between PPI use and CRC [31]; however, its results may also be limited by the aforementioned issues. 

Importantly, ethnicity may contribute to an increased susceptibility to other adverse effects of PPIs, and the Asian population using PPIs is linked to a heightened risk of various severe health conditions [40,41,42]. East Asians, compared to other ethnic groups, are recognized for their slower metabolism of PPIs due to a genetically downregulated expression of hepatic cytochrome p450 enzymes [43], which are essential for PPI metabolism [44]. Most previous studies regarding the association between PPI use and CRC probability were conducted in non-Asian countries [16,17,18,28,29,30,32,33], with only a limited number conducted in Asian populations [31,34,35]. Therefore, performing further validation with national population cohort data that ensure balanced demographics is crucial to diminish the influence of confounding factors and affirm the safety profile of PPIs in Asian populations, in order to better comprehend potential genetic and ethnic variations in their pharmacological impact on CRC. 

We hypothesized that a history of prior PPI exposure and the duration of use could adversely induce the development of CRC and that there might be specific risk factors related to PPI use that could predict CRC occurrence in the Korean population. To test this hypothesis, we conducted a thorough nested case–control study and comprehensive subgroup analyses, carefully matching cases and controls using nationwide public healthcare data.

## 2. Materials and Methods

### 2.1. Participant Selection

In this study, the data were obtained from the Korean National Health Insurance Service–National Sample Cohort (KNHIS-NSC) [33]. The study methods were approved by the Ethics Committee of Hallym University (No. 2022-10-008), and written informed consent was not obligatory as the analysis was performed using anonymous data.

To select eligible participants with CRC, we started with a pool of 1,137,861 individuals with 219,673,817 medical claim codes between 2005 and 2019 (*n* = 9920). The control group comprised individuals not diagnosed with CRC during the same period (*n* = 1,127,941). We excluded participants who were treated for the following conditions using ICD-10 codes: hiatal hernia (K449), gastric surgery (K910–K913), Zollinger–Ellison syndrome (E164), systemic sclerosis (M34), achalasia (K220), and pyloric obstruction (K311–K315), which are known to increase gastrointestinal symptoms or the risk of gastrointestinal cancer [45,46,47,48,49,50]. For the control participants, those later diagnosed with CRC (*n* = 3315) were also excluded.

To minimize selection bias, we performed a 1:4 matching of CRC participants with control participants based on sex, age, income, and region of residence. Control participants were randomly selected during this process. The index date for each CRC participant was recognized as the day when the ICD-10 codes for CRC diagnosis (C18, C19, C20, D010, D011, and D012) were automatically allocated to the participants in the health insurance claims datasets. For control participants, it corresponded to the index date of their matched CRC participant. Therefore, each matched pair shared the same index date. During the matching process, 1,064,426 control participants were excluded. Finally, we successfully matched 9374 CRC participants with 37,496 control participants (Figure 1).

### 2.2. Exposure to PPIs

We conducted a retrospective analysis to determine the period during which participants were prescribed PPIs before being diagnosed with CRC in our cohort groups. Only individuals who commenced PPI use within the year (365 days) leading up to the index date were eligible for our study. We categorized PPI users into two distinct groups based on two criteria: (1) their PPI usage history and (2) the date of their PPI prescription. The history of PPI use was determined based on prescription records and divided into three categories: non-users, current exposure (at least one prescription within the last 29 days before the index date), and past exposure (at least one prescription within the previous 30–365 days before the index date). We calculated the total duration of drug use by considering all prescriptions within the year preceding the index date. Participants were then categorized into four groups: non-users, those with less than 30 days of use, those with 30 to 90 days of use, and those with ≥ 90 days of use.

### 2.3. Outcome (Colorectal Cancer)

To ensure the accuracy of our analysis and reduce the possibility of false-positive cases, we identified individuals with CRC using specific ICD-10 diagnosis codes. These codes encompassed malignant neoplasms of the colon (C18), rectosigmoid junction (C19), rectum (C20), as well as carcinoma in situ of the colon (D010), rectosigmoid junction (D011), and rectum (D012). Within the group of individuals with these diagnosis codes, we chose those with a special claim code for cancer (V193 or V194). The presence of these special claim codes indicated the presence of severe cancer and validated their eligibility for reduced healthcare payments, a policy that has been in place since 2005.

### 2.4. Covariates

Age clusters were stratified into 5-year intervals, ranging from 0–4 years old to ≥85 years, resulting in 18 age clusters. Income clusters were categorized into five classes, from class 1 (lowest income) to class 5 (highest income). The region of residence was clustered into urban and rural areas, following our previous study [51]. To assess the participants’ disease burden, we utilized the Charlson Comorbidity Index (CCI), a widely used tool that considers 17 comorbidities. Each participant received a score based on the severity and number of diseases they had. The CCI was assessed as a continuous variable, ranging from 0 (indicating no comorbidities) to 29 (indicating multiple comorbidities) [52]. However, in this study, cancer was excluded from the CCI score to specifically examine the potential impact of other comorbidities on the development of CRC [52]. Additionally, we evaluated the number of treatments for GERD episodes within 1 year before the index date. Specifically, we focused on individuals treated for GERD (ICD-10 code: K21) at least twice and prescribed a PPI for at least 2 weeks during that time frame. We also recorded the prescription dates of H2-blockers within the year (365 days) preceding the index date. Since PPI users were also more likely to use H2-blockers, we adjusted for the use of H2-blockers as a covariate.

### 2.5. Statistical Analyses

We employed propensity score overlap weighting to account for covariate balance and enhance the effective sample size. The propensity score was calculated using a multivariable logistic regression that included all relevant covariates. In calculating overlap weighting, participants were assigned weights based on the probability of a 1-propensity score for cases and the probability of a propensity score for controls. Overlap weighting values range from 0 to 1 and are designed to achieve exact balance while optimizing the precision [53,54]. We utilized the standardized difference to assess differences in general characteristics between the CRC and control groups. To reduce the possibility of intergroup bias, we evaluated the balance of the matched data regarding the absolute standardized differences in covariates before and after matching. A standardized difference of ≤0.20 indicates a good balance for a particular covariate [55].

Propensity score overlap-weighted multivariable logistic regression for crude (unadjusted) and overlap-weighted (adjusted for age, sex, income, region of residence, CCI, prescription dates of H2-blockers, and the number of treatments for GERD) models were used to estimate the overlap-weighted odds ratios (ORs) and 95% confidence intervals (CIs) for incident CRC regarding the history of PPI use and duration of use by adjusting for potential confounders. Additionally, we conducted subgroup analyses based on age, sex, income, and region of residence. We performed two-tailed analyses, with significance defined as *p*-values below 0.05, utilizing SAS version 9.4 (SAS Institute Inc., Cary, NC, USA).

## 3. Results

This study included 9374 patients with CRC who were matched with a control group of 37,496 individuals from the database. Table 1 displays the baseline demographic characteristics of the participants, both before and after applying overlap weighting adjustments for propensity score matching. Prior to adjustment, there were no notable differences between the CRC and control groups in terms of sex, age, income, residence, CCI score, number of GERD treatments, and H2-blocker prescription dates, as indicated by a standardized difference of less than 0.20. However, the CRC group exhibited higher frequencies of use of PPIs and duration of PPI use compared to the control group, with standardized differences of 0.46 and 0.35, respectively. Following the implementation of overlap-weighting adjustments, demographic distribution attributes between the two groups became more balanced, particularly regarding CCI scores, the number of GERD treatments, and H2-blocker prescription dates. For exposure history and duration of PPI, which still had standardized differences greater than 0.20, additional adjustments were made using logistic regression analysis.

### 3.1. Relationship between the History of PPI Use and CRC Incidence

We investigated the potential link between a history of PPI exposure and the development of CRC, comparing it to a control group (Table 2). Both past and current PPI use were associated with higher odds of developing CRC than in the non-user comparison group ((adjusted OR (aOR) 1.19; 95% CI, 1.12–1.26; *p* < 0.001) and (6.06; 95% CI, 5.60–6.56; *p* < 0.001), respectively).

Subgroup analyses (Figure 2 and Table 3) revealed that the robust association between current PPI use and the likelihood of CRC remained consistent, independent of CCI score, history of H2-blocker use, GERD episodes, income, residential area, sex, or age. Similarly, the relationship between past PPI use and CRC remained consistent, independent of income levels, residential regions, sex, age groups, and with or without GERD episodes.

### 3.2. Relationship between the Duration of PPI Use and Likelihood of CRC 

There was a notable increase in the likelihood of developing CRC, regardless of the overall duration of PPI use in both unadjusted and adjusted models (*p* < 0.001 for all; Table 2). Participants who had been prescribed PPIs for <30 days, 30–90 days, or ≥90 days demonstrated considerably higher odds of developing CRC than the control group (2.71 (95% CI, 2.56–2.87, *p* < 0.001); 1.80 (95% CI, 1.66–1.96, *p* < 0.001); 1.33 (95% CI, 1.19–1.48, *p* < 0.001), respectively). 

In the subgroup analyses (Figure 3 and Table 4), PPI use for either <30 days or 30–90 days remained consistently related to a high likelihood of having CRC, independent of age, sex, income level, or residential area, CCI score, and the presence or absence of GERD or H2-blocker use. The significance observed for the ≥90-day period was consistent across the subgroups, irrespective of age, sex, income, residential area, CCI score ≥1, presence or absence of the use of H2-blockers, or the absence of GERD.

## 4. Discussion

In this nationwide, nested case–control study using well-balanced demographic data and propensity score overlap-weighted multivariable logistic regression analysis, prior PPI use increased the likelihood of CRC independent of whether it was past or present use and the duration of medication, independent of age, sex, income, or residential areas. This seems to favor a possible connection between PPI use and the incidence of CRC in the Korean population. Therefore, our study highlights the necessity to exercise caution when prescribing PPIs to the Korean population to the extent possible, given their widespread usage, in order to mitigate their potential adverse effects.

While large-scale national studies investigating the link between PPI use and CRC risk in Asian populations are limited [31,34,35], a potential link between PPI exposure and incident CRC in three Asian population-based studies was clinically significant. This is particularly noteworthy because most population-based studies conducted in European and US populations have not established an overall connection between PPI usage and CRC risk [16,18,28,29,30]. We found that past and current PPI use was associated with a 1.19- and 6.06-fold higher likelihood of incident CRC compared to the control group (95% CI: 1.12–1.26 and 95% CI: 5.60–6.56, respectively). The odds of developing CRC were significantly higher in individuals exposed to PPIs for less than 30 days, 30–90 days, or over 90 days than in the control group. Our findings align with two Taiwanese and one Korean population-based studies [31,34,35], suggesting a potential association between PPI use and CRC risk. The Taiwanese studies (involving 265 and 3989 CRC cases, respectively), based on the National Health Insurance database, revealed a 2.03- to 2.54-fold higher risk of incident CRC ((95% CI, 1.56–2.63) and (95% CI, 2.31–2.79), respectively) associated with PPI use [34,35]. This increased risk of CRC associated with PPI use was consistent across cumulative use of less than 30 days, 30–90 days, and more than 90 days [34]. 

The Korean prospective cohort study, which included 5304 CRC patients aged ≥ 40 years and utilized Korean public health insurance data from 2002 to 2006, identified an association between PPI use and CRC risk in specific sub-populations with low-risk factors for CRC [31]. Non-obese, non-diabetic females below 50 years with no history of alcohol consumption who received ≥180 days of cumulative PPI use had a 12.30-fold higher risk of CRC (95% CI, 1.71–88.23) compared to that of those who did not use PPIs [31]. However, their study focused exclusively on middle-aged and older populations [31], making it impossible to determine whether PPI usage might increase the risk of CRC in younger individuals, particularly due to the increasing incidence of CRC in this population group in Korea [3]. Our investigation utilized a more up-to-date database with a broader timeframe (2005–2019) and included individuals of all ages while also adjusting for the use of H2-blockers as covariates in the analysis. In our study, we observed an elevated occurrence of CRC independent of prior PPI history or the duration of PPI usage, and this trend remained consistent across all age groups, both sexes, different income levels, various residential areas, and regardless of the use of H2-blockers or GERD history. These findings may suggest that in a Korean setting, PPI use may be an independent risk factor for incident CRC. 

The findings of the European studies [29,32] suggesting a potential connection between current PPI use and CRC likelihood may also support our findings. In a Dutch study, which relied on data from the Netherlands Cancer Registry and the General Practitioner Database (2007–2014), current PPI use was associated with a 1.30-fold greater risk of CRC (95% CI, 1.16–1.47) [32]. In a nested case–control study among patients ≥ 50 years in the UK, recent PPI use within 1 year showed a 2.6-times stronger association with the risk of CRC (95% CI, 2.3–2.9) [29]. The studies alluded to the possibility of protopathic bias or the outcome of reverse causality [29,32], as the association between PPI use and CRC risk persisted for recent PPI use but not for past use [29]. However, in this study, we found that both past and current PPI users had an increased likelihood of developing CRC, even after adjusting for GERD episodes and H2-blocker use and excluding diseases that cause gastrointestinal symptoms. PPI exposure for less than 3 months before a CRC diagnosis may not align with the traditional hypothesis of the adenoma–carcinoma sequence [29,32,35], which typically involves much longer dwelling times, such as 26 years for tubular adenoma, 9 years for tubulovillous adenoma, and 4 years for villous adenoma [56]. Nevertheless, de novo CRC, which arises without a prior adenomatous stage from normal mucosa, accounts for a significant percentage of cases in Asia, ranging from 20% to 90% [56,57,58], and exhibits highly invasive and metastatic characteristics, often presenting initially with systemic metastasis and rapidly progressing, with a tumor doubling time of approximately 3 months [59,60]. Similarly, in a mouse model, PPI-administered mice for 4 weeks showed colon cancer growth [27]. Our findings seem to align with these explanations.

Indeed, PPIs can induce secondary hypergastrinemia, which in certain PPI-treated individuals can lead to significantly elevated plasma gastrin levels and a heightened risk of high-grade dysplasia or CRC [22]. The adverse effects of current PPI exposure on incident CRC might be attributed to the irreversible binding of PPIs to proton pumps in gastric parietal cells, resulting in prolonged effects on gastric pH levels [10,12]. This, in turn, can impact the gut microbiome [24,25], and even short-term exposure to PPIs may potentially contribute to the development of gastrointestinal tumors [26]. Additionally, since PPIs are metabolized by hepatic cytochrome P450 [43], individuals with genetic polymorphisms commonly found in some Asian populations, leading to slower metabolism, may attain higher levels of PPIs even with short-term and relatively low-dose use [43,61]. In contrast, H2-blockers, which only block the effects of histamine, are less likely to induce hypergastrinemia and may be less related to an increased risk of CRC from a theoretical perspective. This could partially explain why the use of PPIs, rather than H2-blockers, was associated with an elevated likelihood of incident CRC in our study. 

The strength of this study is that the results were drawn from a large cohort representing the entire Korean population. The NHIS-NSC database provides extensive access to participants’ medical histories from healthcare facilities nationwide, thus improving the generalizability and accuracy of the study outcomes. Through overlap-weighted propensity score matching, our study successfully achieved a balanced distribution of sex, age, income, residence, and CCI score among the participants, which effectively minimized selection bias and created study groups that closely resembled those observed in randomized clinical trials [62]. To the best of our knowledge, in many countries where epidemiological investigations have been carried out, PPIs are available as over-the-counter drugs that can be obtained at pharmacies without the need for a doctor’s prescription, with the exception of South Korea and Taiwan [63]. The accessibility of and regulations surrounding the availability of PPIs can vary significantly between countries, and these differences may impact the patterns of PPI use and their potential effects. Given the variations in drug accessibility and healthcare practices across different countries, it is essential to consider how these factors might influence the relationship between PPI use and CRC risk. Our study in South Korea, where PPIs are prescription-based, provides a unique perspective within this specific healthcare context. In a recent study conducted in Taiwan, it was found that CRC patients who concurrently used PPIs with chemotherapy for more than 60 days faced a 1.10-fold elevated risk of cancer-specific death (95% CI, 1.01–1.17) [63], which underscores the potentially detrimental significance of PPI usage in CRC patients during cancer treatment. By examining the association between PPI use and CRC risk in a setting where these drugs are available exclusively by prescription, our study evaluating the impact of prior PPI usage history on the development of CRC in the Korean population may contribute to a more comprehensive understanding of the multifaceted interactions between PPIs and CRC. 

This study had some limitations. Given that this study solely concentrated on Korean citizens, extrapolating the findings to wider populations, especially those in regions where PPIs are available as over-the-counter medications obtainable at pharmacies without a prescription, may pose challenges. Ethnicity might also play a strong role in the effect of PPIs on CRC incidence. While we adjusted for variables associated with PPI use to minimize confounding effects, the retrospective design and the reliance on diagnosis codes from Korean health insurance data, as well as the lack of information regarding CRC stage, location, phenotype, and genetic characteristics, may introduce limitations on our findings and potentially result in unmeasured confounding effects, thus making it difficult to apply our results to other demographic groups.

## 5. Conclusions

Our findings carefully suggest that prior PPI use, regardless of past or present use and irrespective of duration of use, may be associated with an increased risk of developing CRC in the Korean population and may necessitate the need for further information and education on incident CRC as a rare PPI-related adverse effect.

## Figures and Tables

**Figure 1 cancers-15-05606-f001:**
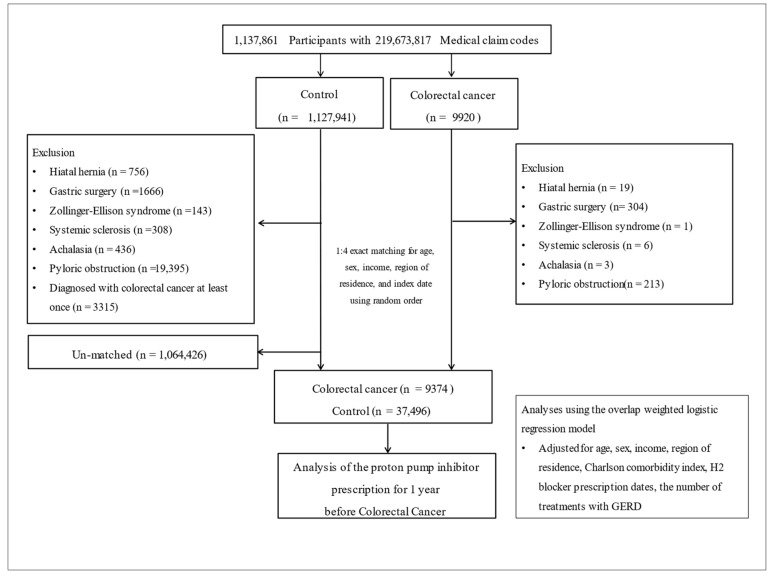
Schematic diagram of the participant selection process used in this study. Of 1,137,861 participants, 9374 participants with colorectal cancer were matched with 37,496 control participants for sex, age, income, and region of residence.

**Figure 2 cancers-15-05606-f002:**
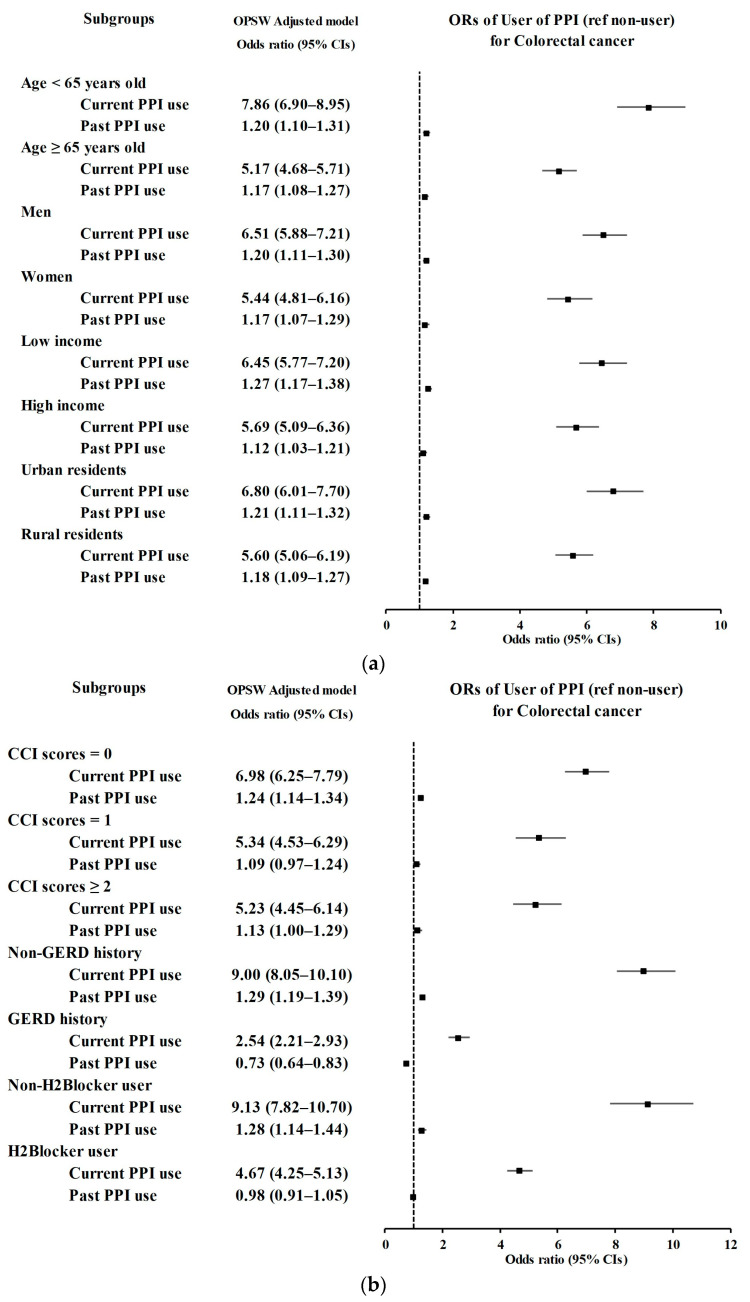
Subgroup analyses of PPI users (non-user [ref] vs. user) for colorectal cancer according to (**a**) age, sex, income, and region of residence and (**b**) CCI score, prescription dates of H2-blockers, and the number of GERD treatments visualized on a forest plot. Abbreviations: PPI, proton pump inhibitor; CCI, Charlson Comorbidity Index.

**Figure 3 cancers-15-05606-f003:**
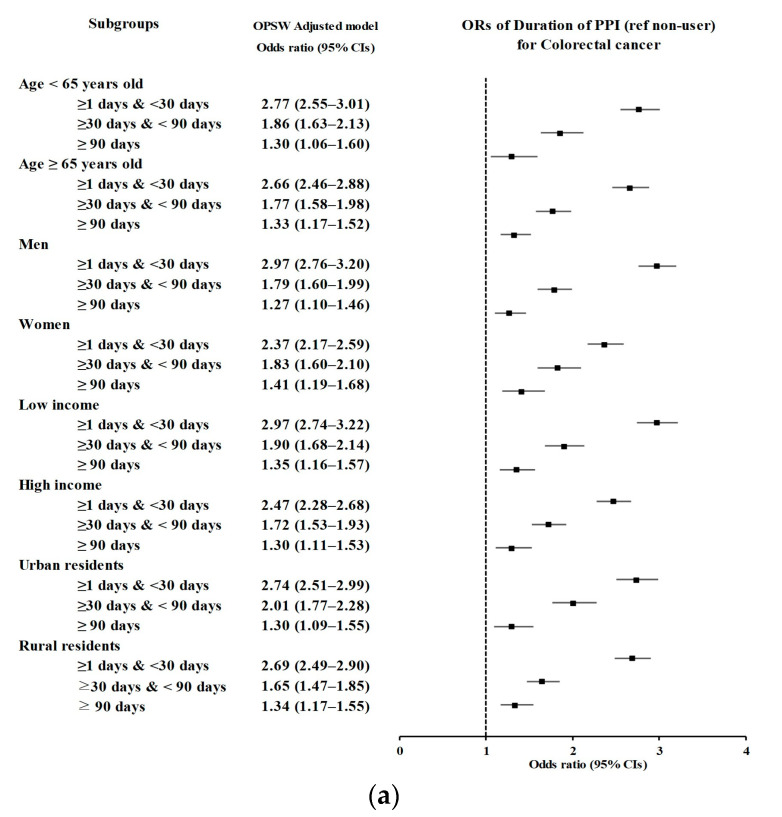
Subgroup analyses of PPI duration (non-user (ref) vs. ≥1 day and <30 days, ≥30 days and < 90 days, and ≥90 days PPI prescription dates) for colorectal cancer according to (**a**) age, sex, income, and region of residence and (**b**) CCI score, prescription dates of H2-blockers, and number of GERD treatments visualized on forest plots. Abbreviations: PPI, proton pump inhibitor; CCI, Charlson Comorbidity Index.

**Table 1 cancers-15-05606-t001:** General characteristics of participants.

Characteristics		Before PS Overlap Weighting Adjustment			After PS Overlap Weighting Adjustment		
		Colorectal Cancer	Control	Standardized Difference	Colorectal Cancer	Control	Standardized Difference
Age (%)				0.00			0.00
	0–4	1 (0.01)	4 (0.01)		1 (0.01)	1 (0.01)	
	5–9	N/A	N/A		N/A	N/A	
	10–14	3 (0.03)	12 (0.03)		2 (0.03)	2 (0.03)	
	15–19	1 (0.01)	4 (0.01)		1 (0.01)	1 (0.01)	
	20–24	8 (0.09)	32 (0.09)		6 (0.09)	6 (0.09)	
	25–29	26 (0.28)	104 (0.28)		21 (0.28)	21 (0.28)	
	30–34	88 (0.94)	352 (0.94)		70 (0.94)	70 (0.94)	
	35–39	174 (1.86)	696 (1.86)		139 (1.85)	139 (1.85)	
	40–44	346 (3.69)	1384 (3.69)		276 (3.68)	276 (3.68)	
	45–49	544 (5.80)	2176 (5.80)		434 (5.80)	434 (5.80)	
	50–54	926 (9.88)	3704 (9.88)		739 (9.87)	739 (9.87)	
	55–59	1183 (12.62)	4732 (12.62)		945 (12.62)	945 (12.62)	
	60–64	1321 (14.09)	5284 (14.09)		1054 (14.07)	1054 (14.07)	
	65–69	1396 (14.89)	5584 (14.89)		1115 (14.89)	1115 (14.89)	
	70–74	1383 (14.75)	5532 (14.75)		1104 (14.75)	1104 (14.75)	
	75–79	987 (10.53)	3948 (10.53)		790 (10.54)	790 (10.54)	
	80–84	626 (6.68)	2504 (6.68)		502 (6.71)	502 (6.71)	
	85+	361 (3.85)	1444 (3.85)		289 (3.86)	289 (3.86)	
Sex (%)				0.00			0.00
	Male	5596 (59.70)	22,384 (59.70)		4469 (59.68)	4469 (59.68)	
	Female	3778 (40.30)	15,112 (40.30)		3019 (40.32)	3019 (40.32)	
Income (%)				0.00			0.00
	1 (lowest)	1861 (19.85)	7444 (19.85)		1487 (19.85)	1487 (19.85)	
	2	1196 (12.76)	4784 (12.76)		954 (12.75)	954 (12.75)	
	3	1480 (15.79)	5920 (15.79)		1181 (15.77)	1181 (15.77)	
	4	1954 (20.84)	7816 (20.84)		1562 (20.86)	1562 (20.86)	
	5 (highest)	2883 (30.76)	11,532 (30.76)		2304 (30.77)	2304 (30.77)	
Region of residence (%)				0.00			0.00
	Urban	4220 (45.02)	16,880 (45.02)		3370 (45.01)	3370 (45.01)	
	Rural	5154 (54.98)	20,616 (54.98)		4118 (54.99)	4118 (54.99)	
CCI score (Mean, SD)		0.79 (1.17)	0.79 (1.17)	0.09	0.77 (1.03)	0.77 (0.57)	0.00
Number of treatments with GERD (Mean, SD)		0.49 (1.74)	0.49 (1.74)	0.03	0.48 (1.51)	0.48 (0.95)	0.00
H2-blocker prescription dates (Mean, SD)		30.08 (62.38)	30.08 (62.38)	0.05	29.58 (54.96)	29.59 (31.48)	0.00
PPI users (*n*, %)				0.46			0.46
	Non-use	6607 (70.48)	31,362 (83.64)		5287 (70.61)	6221 (83.08)	
	Current PPI use	1666 (17.77)	1401 (3.74)		1325 (17.70)	295 (3.93)	
	Past PPI use	1101 (11.75)	4733 (12.62)		875 (11.69)	972 (12.99)	
Duration of PPI use (*n*, %)				0.35			0.33
	Non-use	6607 (70.48)	31,362 (83.64)		5287 (70.61)	6221 (83.08)	
	≥1 days and <30 days	1839 (19.62)	3249 (8.66)		1468 (19.60)	659 (8.81)	
	≥30 days and <90 days	606 (6.46)	1649 (4.40)		481 (6.42)	343 (4.58)	
	≥90 days	322 (3.44)	1236 (3.30)		252 (3.37)	265 (3.54)	

Abbreviations: CCI, Charlson Comorbidity Index; PS, propensity score; GERD, gastroesophageal reflux disease; N/A, not applicable.

**Table 2 cancers-15-05606-t002:** Crude and overlap propensity score weighted odd ratios of proton pump inhibitors (ref: non-user) for colorectal cancer.

Characteristics	Colorectal Cancer	Control	Odds Ratios (95% Confidence Intervals)			
	(Exposure/Total, %)	(Exposure/Total, %)	Crude	*p*	Adjusted Model with OW †	*p*
Exposure to PPI						
Past	1101/9374 (11.75)	4733/37,496 (12.62)	1.10 (1.03–1.19)	0.006 *	1.19 (1.12–1.26)	<0.001 *
Current	1666/9374 (17.77)	1401/37,496 (3.74)	5.64 (5.23–6.09)	<0.001 *	6.06 (5.60–6.56)	<0.001 *
Duration of PPI use						
<30 days	1839/9374 (19.62)	3249/37,496 (8.66)	2.69 (2.52–2.86)	<0.001 *	2.71 (2.56–2.87)	<0.001 *
30–90 days	606/9374 (6.46)	1649/37,496 (4.4)	1.74 (1.58–1.92)	<0.001 *	1.80 (1.66–1.96)	<0.001 *
≥90 days	322/9374 (3.44)	1236/37,496 (3.3)	1.24 (1.09–1.40)	<0.001 *	1.33 (1.19–1.48)	<0.001 *

Abbreviations: PPI, proton pump inhibitor; OW, overlap propensity score weighted adjustment. * Logistic regression model, significance at *p* < 0.05. † Adjusted for age, sex, income, region of residence, Charlson Comorbidity Index scores, the number of gastroesophageal reflux disease treatments, and prescription dates of H_2_-receptor antagonist.

**Table 3 cancers-15-05606-t003:** Subgroup analyses regarding odds ratio (95% confidence intervals) of PPI users for colorectal cancer.

User of PPI		Colorectal Cancer	Control	Odds Ratios (95% Confidence Intervals)			
		(Exposure/Total, %)	(Exposure/Total, %)	Crude	*p*	Adjusted Model with OW †	*p*
Age < 65 years old (*n* = 23,105)							
	Current PPI use	738/4621 (15.97)	482/18,484 (2.61)	7.19 (6.38–8.12)	<0.001 *	7.86 (6.90–8.95)	<0.001 *
	Past PPI use	488/4621 (10.56)	2052/18,484 (11.1)	1.12 (1.01–1.24)	0.039 *	1.20 (1.10–1.31)	<0.001 *
Age ≥ 65 years old (*n* = 23,765)							
	Current PPI use	928/4753 (19.52)	919/19,012 (4.83)	4.85 (4.39–5.35)	<0.001 *	5.17 (4.68–5.71)	<0.001 *
	Past PPI use	613/4753 (12.9)	2681/19,012 (14.1)	1.10 (1.00–1.21)	0.058	1.17 (1.08–1.27)	<0.001 *
Males (*n* = 27,980)							
	Current PPI use	1040/5596 (18.58)	820/22,384 (3.66)	6.09 (5.52–6.71)	<0.001 *	6.51 (5.88–7.21)	<0.001 *
	Past PPI use	645/5596 (11.53)	2783/22,384 (12.43)	1.11 (1.01–1.22)	0.023 *	1.20 (1.11–1.30)	<0.001 *
Females (*n* = 18,890)							
	Current PPI use	626/3778 (16.57)	581/15,112 (3.84)	5.03 (4.46–5.67)	<0.001 *	5.44 (4.81–6.16)	<0.001 *
	Past PPI use	456/3778 (12.07)	1950/15,112 (12.9)	1.09 (0.98–1.22)	0.12	1.17 (1.07–1.29)	<0.001 *
Low income (*n* = 22,685)							
	Current PPI use	868/4537 (19.13)	711/18148 (3.92)	5.93 (5.33–6.59)	<0.001 *	6.45 (5.77–7.20)	<0.001 *
	Past PPI use	554/4537 (12.21)	2310/18148 (12.73)	1.16 (1.05–1.29)	0.003 *	1.27 (1.17–1.38)	<0.001 *
High income (*n* = 24,185)							
	Current PPI use	798/4837 (16.5)	690/19,348 (3.57)	5.38 (4.82–5.99)	<0.001 *	5.69 (5.09–6.36)	<0.001 *
	Past PPI use	547/4837 (11.31)	2423/19,348 (12.52)	1.05 (0.95–1.16)	0.342	1.12 (1.03–1.21)	0.008 *
Urban (*n* = 21,100)							
	Current PPI use	721/4220 (17.09)	543/16,880 (3.22)	6.29 (5.59–7.08)	<0.001 *	6.80 (6.01–7.70)	<0.001 *
	Past PPI use	476/4220 (11.28)	2016/16,880 (11.94)	1.12 (1.00–1.25)	0.041 *	1.21 (1.11–1.32)	<0.001 *
Rural (*n* = 25,770)							
	Current PPI use	945/5154 (18.34)	858/20,616 (4.16)	5.24 (4.74–5.78)	<0.001 *	5.60 (5.06–6.19)	<0.001 *
	Past PPI use	625/5154 (12.13)	2717/20,616 (13.18)	1.09 (1.00–1.20)	0.062	1.18 (1.09–1.27)	<0.001 *
CCI scores = 0 (*n* = 28,945)							
	Current PPI use	823/5175 (15.9)	705/23,770 (2.97)	6.28 (5.64–6.98)	<0.001 *	6.98 (6.25–7.79)	<0.001 *
	Past PPI use	559/5175 (10.8)	2667/23,770 (11.22)	1.13 (1.02–1.24)	0.016 *	1.24 (1.14–1.34)	<0.001 *
CCI scores = 1 (*n* = 9926)							
	Current PPI use	456/2453 (18.59)	341/7473 (4.56)	4.79 (4.12–5.56)	<0.001 *	5.34 (4.53–6.29)	<0.001 *
	Past PPI use	300/2453 (12.23)	1058/7473 (14.16)	1.01 (0.88–1.17)	0.835	1.09 (0.97–1.24)	0.161
CCI scores ≥ 2 (*n* = 7999)							
	Current PPI use	387/1746 (22.16)	355/6253 (5.68)	4.77 (4.07–5.59)	<0.001 *	5.23 (4.45–6.14)	<0.001 *
	Past PPI use	242/1746 (13.86)	1008/6253 (16.12)	1.05 (0.90–1.23)	0.528	1.13 (1.00–1.29)	0.058
Without GERD (*n* = 40,708)							
	Current PPI use	1071/7918 (13.53)	557/32,790 (1.7)	9.24 (8.31–10.3)	<0.001 *	9.00 (8.05–10.1)	<0.001 *
	Past PPI use	594/7918 (7.5)	2186/32,790 (6.67)	1.31 (1.19–1.44)	<0.001 *	1.29 (1.19–1.39)	<0.001 *
With GERD (*n* = 6162)							
	Current PPI use	595/1456 (40.87)	844/4706 (17.93)	2.62 (2.24–3.07)	<0.001 *	2.54 (2.21–2.93)	<0.001 *
	Past PPI use	507/1456 (34.82)	2547/4706 (54.12)	0.74 (0.64–0.86)	<0.001 *	0.73 (0.64–0.83)	<0.001 *
Without H2-blocker use (*n* = 19191)							
	Current PPI use	314/2281 (13.77)	314/16,910 (1.86)	8.57 (7.27–10.1)	<0.001 *	9.13 (7.82–10.7)	<0.001 *
	Past PPI use	171/2281 (7.5)	1209/16,910 (7.15)	1.21 (1.02–1.43)	0.025 *	1.28 (1.14–1.44)	<0.001 *
With H2-blocker use (*n* = 27,679)							
	Current PPI use	1352/7093 (19.06)	1087/20,586 (5.28)	4.13 (3.79–4.50)	<0.001 *	4.67 (4.25–5.13)	<0.001 *
	Past PPI use	930/7093 (13.11)	3524/20,586 (17.12)	0.88 (0.81–0.95)	0.001 *	0.98 (0.91–1.05)	0.613

Abbreviations: CCI, Charlson comorbidity index; GERD, gastroesophageal reflux disease; OW, overlap weighting; PPI, proton pump inhibitor; * Logistic regression model, significance at *p* < 0.05; † Adjusted for age, sex, income, region of residence, CCI score, H2-blocker dates, and the number of GERD treatments.

**Table 4 cancers-15-05606-t004:** Subgroup analyses regarding odds ratio (95% confidence intervals) of duration of PPI for colorectal cancer.

Duration of PPI		Colorectal Cancer	Control	Odds Ratios (95% Confidence Intervals)			
		(Exposure/Total, %)	(Exposure/Total, %)	Crude	*p*	Adjusted Model with OW †	*p*
Age < 65 years old (*n* = 23,105)							
	<30 days	872/4621 (18.87)	1499/18,484 (8.11)	2.73 (2.49–2.99)	<0.001 *	2.77 (2.55–3.01)	<0.001 *
	30–90 days	261/4621 (5.65)	687/18,484 (3.72)	1.78 (1.54–2.07)	<0.001 *	1.86 (1.63–2.13)	<0.001 *
	≥90 days	93/4621 (2.01)	348/18,484 (1.88)	1.26 (1.00–1.58)	0.054	1.30 (1.06–1.60)	0.012 *
Age ≥ 65 years old (*n* = 23,765)							
	<30 days	967/4753 (20.35)	1750/19,012 (9.2)	2.65 (2.43–2.89)	<0.001 *	2.66 (2.46–2.88)	<0.001 *
	30–90 days	345/4753 (7.26)	962/19,012 (5.06)	1.72 (1.51–1.96)	<0.001 *	1.77 (1.58–1.98)	<0.001 *
	≥90 days	229/4753 (4.82)	888/19,012 (4.67)	1.24 (1.06–1.44)	0.005 *	1.33 (1.17–1.52)	<0.001 *
Males (*n* = 27,980)							
	<30 days	1128/5596 (20.16)	1836/22,384 (8.2)	2.95 (2.72–3.20)	<0.001 *	2.97 (2.76–3.20)	<0.001 *
	30–90 days	371/5596 (6.63)	1025/22,384 (4.58)	1.74 (1.54–1.97)	<0.001 *	1.79 (1.60–1.99)	<0.001 *
	≥90 days	186/5596 (3.32)	742/22,384 (3.31)	1.20 (1.02–1.42)	0.027 *	1.27 (1.10–1.46)	0.001 *
Females (*n* = 18,890)							
	<30 days	711/3778 (18.82)	1413/15,112 (9.35)	2.35 (2.13–2.59)	<0.001 *	2.37 (2.17–2.59)	<0.001 *
	30–90 days	235/3778 (6.22)	624/15,112 (4.13)	1.76 (1.50–2.05)	<0.001 *	1.83 (1.60–2.10)	<0.001 *
	≥90 days	136/3778 (3.6)	494/15,112 (3.27)	1.28 (1.06–1.56)	0.011 *	1.41 (1.19–1.68)	<0.001 *
Low income (*n* = 22,685)							
	<30 days	953/4537 (21.01)	1570/18,148 (8.65)	2.95 (2.70–3.22)	<0.001 *	2.97 (2.74–3.22)	<0.001 *
	30–90 days	298/4537 (6.57)	790/18,148 (4.35)	1.83 (1.59–2.10)	<0.001 *	1.90 (1.68–2.14)	<0.001 *
	≥90 days	171/4537 (3.77)	661/18,148 (3.64)	1.26 (1.06–1.49)	0.01 *	1.35 (1.16–1.57)	<0.001 *
High income (*n* = 24,185)							
	<30 days	886/4837 (18.32)	1679/19,348 (8.68)	2.45 (2.24–2.68)	<0.001 *	2.47 (2.28–2.68)	<0.001 *
	30–90 days	308/4837 (6.37)	859/19,348 (4.44)	1.67 (1.46–1.91)	<0.001 *	1.72 (1.53–1.93)	<0.001 *
	≥90 days	151/4837 (3.12)	575/19,348 (2.97)	1.22 (1.02–1.47)	0.032 *	1.30 (1.11–1.53)	0.001 *
Urban (*n* = 21,100)							
	<30 days	788/4220 (18.67)	1378/16,880 (8.16)	2.71 (2.46–2.98)	<0.001 *	2.74 (2.51–2.99)	<0.001 *
	30–90 days	289/4220 (6.85)	708/16,880 (4.19)	1.93 (1.68–2.23)	<0.001 *	2.01 (1.77–2.28)	<0.001 *
	≥90 days	120/4220 (2.84)	473/16,880 (2.8)	1.20 (0.98–1.47)	0.077	1.30 (1.09–1.55)	0.004 *
Rural (*n* = 25,770)							
	<30 days	1051/5154 (20.39)	1871/20,616 (9.08)	2.67 (2.46–2.90)	<0.001 *	2.69 (2.49–2.90)	<0.001 *
	30–90 days	317/5154 (6.15)	941/20,616 (4.56)	1.60 (1.40–1.83)	<0.001 *	1.65 (1.47–1.85)	<0.001 *
	≥90 days	202/5154 (3.92)	763/20,616 (3.7)	1.26 (1.07–1.48)	0.005 *	1.34 (1.17–1.55)	<0.001 *
CCI scores = 0 (*n* = 28,945)							
	<30 days	981/5175 (18.96)	1930/23,770 (8.12)	2.73 (2.51–2.97)	<0.001 *	2.76 (2.56–2.97)	<0.001 *
	30–90 days	305/5175 (5.89)	895/23,770 (3.77)	1.83 (1.60–2.10)	<0.001 *	1.91 (1.70–2.14)	<0.001 *
	≥90 days	96/5175 (1.86)	547/23,770 (2.3)	0.94 (0.76–1.18)	0.606	1.05 (0.87–1.25)	0.627
CCI scores = 1 (*n* = 9926)							
	<30 days	500/2453 (20.38)	719/7473 (9.62)	2.49 (2.19–2.82)	<0.001 *	2.55 (2.26–2.89)	<0.001 *
	30–90 days	158/2453 (6.44)	384/7473 (5.14)	1.47 (1.21–1.79)	<0.001 *	1.54 (1.29–1.85)	<0.001 *
	≥90 days	98/2453 (4)	296/7473 (3.96)	1.19 (0.94–1.50)	0.156	1.37 (1.10–1.72)	0.006 *
CCI scores ≥ 2 (*n* = 7999)							
	<30 days	358/1746 (20.5)	600/6253 (9.6)	2.61 (2.26–3.02)	<0.001 *	2.72 (2.38–3.11)	<0.001 *
	30–90 days	143/1746 (8.19)	370/6253 (5.92)	1.69 (1.38–2.07)	<0.001 *	1.76 (1.47–2.11)	<0.001 *
	≥90 days	128/1746 (7.33)	393/6253 (6.28)	1.43 (1.16–1.76)	<0.001 *	1.56 (1.29–1.89)	<0.001 *
Without GERD (*n* = 40,708)							
	<30 days	1288/7918 (16.27)	1699/32,790 (5.18)	3.64 (3.37–3.94)	<0.001 *	3.61 (3.36–3.88)	<0.001 *
	30–90 days	257/7918 (3.25)	591/32,790 (1.8)	2.09 (1.80–2.43)	<0.001 *	2.02 (1.78–2.30)	<0.001 *
	≥90 days	120/7918 (1.52)	453/32,790 (1.38)	1.27 (1.04–1.56)	0.02 *	1.22 (1.04–1.43)	0.017 *
With GERD (*n* = 6162)							
	<30 days	551/1456 (37.84)	1550/4706 (32.94)	1.32 (1.13–1.54)	<0.001 *	1.32 (1.16–1.50)	<0.001 *
	30–90 days	349/1456 (23.97)	1058/4706 (22.48)	1.23 (1.04–1.45)	0.018 *	1.20 (1.04–1.38)	0.013 *
	≥90 days	202/1456 (13.87)	783/4706 (16.64)	0.96 (0.79–1.16)	0.667	0.92 (0.78–1.08)	0.288
Without H2-blocker use (*n* = 19,191)							
	<30 days	334/2281 (14.64)	831/16,910 (4.91)	3.44 (3.01–3.95)	<0.001 *	3.43 (3.08–3.82)	<0.001 *
	30–90 days	101/2281 (4.43)	397/16,910 (2.35)	2.18 (1.74–2.73)	<0.001 *	2.06 (1.74–2.45)	<0.001 *
	≥90 days	50/2281 (2.19)	295/16,910 (1.74)	1.45 (1.07–1.97)	0.016 *	1.31 (1.03–1.67)	0.029 *
With H2-blocker use (*n* = 26,679)							
	<30 days	1505/7093 (21.22)	2418/20,586 (11.75)	2.07 (1.92–2.22)	<0.001 *	2.13 (1.99–2.29)	<0.001 *
	30–90 days	505/7093 (7.12)	1252/20,586 (6.08)	1.34 (1.20–1.49)	<0.001 *	1.47 (1.33–1.63)	<0.001 *
	≥90 days	272/7093 (3.83)	941/20,586 (4.57)	0.96 (0.84–1.10)	0.562	1.17 (1.03–1.34)	0.018 *

Abbreviations: CCI, Charlson Comorbidity Index; GERD, gastroesophageal reflux disease; OW, overlap weighting; PPI, proton pump inhibitor; * Logistic regression model, Significance at *p* < 0.05; † Adjusted for age, sex, income, region of residence, CCI score, H2-blocker dates, and the number of GERD treatment.

## Data Availability

Restrictions apply to the availability of these data. Data were obtained from the Korean National Health Insurance Sharing Service (NHISS) and are available at https://nhiss.nhis.or.kr (accessed on 25 January 2022) with the permission of the NHIS.

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
