# Peer review of "Proton Pump Inhibitors and Likelihood of Colorectal Cancer in the Korean Population: Insights from a Nested Case–Control Study Using National Health Insurance Data"

_cancers, 2023, doi:10.3390/cancers15235606_

Round 1
Reviewer 1 Report
Comments and Suggestions for Authors
Form the technical and methodological point of view, presented paper was well designed, and obtained results were correctly elaborated. However, obtained data have limited novelty to me, as we considered that there are many studies of such type, including recently published study performed in Taiwan on the same medical problem (https://www.mdpi.com/2072-6694/15/21/5304). This study is almost the same and was published in the same Journal, less than a month ago. The only difference between this two studies is that the data in the present paper are for Korean population. The rest is almost the same and conclusions are very similar. In addition to this major complaint, I described below some minor inaccuracies that should be corrected:
1. line 54 - what did the Authors mean "considered a low-risk area". Low-risk in the term of low exposure to some contaimants? By air, food? Please explain it in the text.
2. Introduction - the Authors wrote only that PPIs are prescribed by doctors for specific health conditions. However, in Europe these drugs since around 10 years are also widely available as OTC drugs. Please describe what is the situation in Korea.
3. line 110 - Authors should also refer to the recent study performed in Taiwan, which I have mentioned above.
4. line 289-290 - as a very similar study was recently published in Taiwan, Authors can't say the similar studies are limited in Asian countries. Please modify this part, including a reference to a mentioned publication
5. line 366 - in my opinion additional limitation of the study is lack of data for patients, which have been used PPIs as OTC drugs, of course in the situation when it is possible in Korea. Therefore, in the Introduction it must be clearly written if such drugs are available as OTC, and a brief description in what regions of the world PPIs are available as OTC drugs.
Author Response
Reviewer #1:
General Comments: Form the technical and methodological point of view, presented paper was well designed, and obtained results were correctly elaborated. However, obtained data have limited novelty to me, as we considered that there are many studies of such type, including recently published study performed in Taiwan on the same medical problem (https://www.mdpi.com/2072-6694/15/21/5304). This study is almost the same and was published in the same Journal, less than a month ago. The only difference between this two studies is that the data in the present paper are for Korean population. The rest is almost the same and conclusions are very similar. In addition to this major complaint, I described below some minor inaccuracies that should be corrected:
Response: We would like to express our sincere gratitude to the reviewer for dedicating their valuable time and expertise to the thorough assessment of our manuscript. We truly appreciate your insightful feedback, and we are committed to addressing your concerns and improving our work.
We wish to clarify that our research commenced in May 2023 with the support of research funding (HURF-2023-21). Unfortunately, we were not aware of the recent Taiwanese publication at the time of submitting our study. We sincerely apologize for any unintentional overlap in research topics.
While we understand your concern, we believe that the inclusion of data from different geographical regions can contribute to a more comprehensive understanding of the relationship between PPIs and colorectal cancer, particularly considering the variability in healthcare practices and drug accessibility in different countries.
Once again, we appreciate your thoughtful comments, and we are committed to ensuring the quality and relevance of our research. Your insights have been invaluable in helping us refine our work.
Comment 1: Line 54 - what did the Authors mean "considered a low-risk area". Low-risk in the term of low exposure to some contaimants? By air, food? Please explain it in the text.
Response: We sincerely appreciate the reviewer's question regarding the term "low-risk area." We originally used this term to describe a region with a historically low incidence of colorectal cancer that has undergone a significant transformation over the years. Specifically, between 1987 and 1996, the age-standardized mortality rates increased from 8.7 in 100,000 to 16.5 in 100,000 for men and from 6.3 in 100,000 to 14.3 in 100,000 for women in South Korea. Furthermore, by 2018, the estimated incidence rate had surged to 44.5 cases per 100,000 persons per year, marking a substantial rise in colorectal cancer cases. This transformation is not unique to South Korea but is part of a global trend. In 2018, South Korea ranked second in the world for colorectal cancer incidence rates. In the country, cancer has emerged as the leading cause of death, with colorectal cancer ranking third in mortality after lung and liver cancer.
To avoid confusion, we have removed the term “low-risk area” from the revised manuscript and more clearly explained the facts stated above:
(Introduction, lines 54-57): In South Korea, the epidemiology of CRC has been undergoing significant transformations for recent decades, with the nation witnessing the world's second-highest incidence rate of CRC in 2018, at 44.5 cases per 100000 individuals annually.
Comment 2: Introduction - the Authors wrote only that PPIs are prescribed by doctors for specific health conditions. However, in Europe these drugs since around 10 years are also widely available as OTC drugs. Please describe what is the situation in Korea.
Response: We appreciate the reviewer's comment regarding the availability of PPIs as over-the-counter (OTC) drugs in Europe. However, we would like to clarify that PPIs are not available as OTC drugs in South Korea. The prescription of PPIs in South Korea requires a doctor's authorization.
We understand that variations in drug accessibility and healthcare practices across different countries can influence the use and potential effects of PPIs. Therefore, we believe that publishing data and results from South Korea, where PPIs are prescription-based, can offer valuable insights into the relationship between PPI use and colorectal cancer risk within this specific healthcare context.
We have further clarified this in the revised manuscript:
(Introduction, lines 67-68): Although PPIs are not an over-the-count drug in South Korea, their usage has increased more than 1.5 times over the past five years.
Comment 3: line 110 - Authors should also refer to the recent study performed in Taiwan, which I have mentioned above.
Response: Thank you for bringing this study to our attention. We have incorporated it and highlighted its significance and distinctions from our study in the Discussion section, as follows:
(Discussion, lines 368-385): To the best of our knowledge, in many countries where epidemiological investigations have been carried out, PPIs are available as over-the-counter drugs that can be obtained at pharmacies without the need for a doctor's prescription, with the exception of South Korea and Taiwan. The accessibility of and regulations surrounding the availability of PPIs can vary significantly between countries, and these differences may impact the patterns of PPI use and their potential effects. Given the variations in drug accessibility and healthcare practices across different countries, it is essential to consider how these factors might influence the relationship between PPI use and CRC risk. Our study in South Korea, where PPIs are prescription-based, provides a unique perspective within this specific healthcare context. In a recent study conducted in Taiwan, it was found that CRC patients who concurrently used PPIs with chemotherapy for more than 60 days faced a 1.10-fold elevated risk of cancer-specific death (95% CI, 1.01–1.17) [63], which underscores the potentially detrimental significance of PPI usage in CRC patients during cancer treatment. By examining the association between PPI use and CRC risk in a setting where these drugs are available exclusively by prescription, our study evaluating the impact of prior PPI usage history on the development of CRC in the Korean population may contribute to a more comprehensive understanding of the multifaceted interactions between PPIs and CRC.
Comment 4: line 289-290 - as a very similar study was recently published in Taiwan, Authors can't say the similar studies are limited in Asian countries. Please modify this part, including a reference to a mentioned publication.
Response: We genuinely appreciate the reviewer's input and the reference to the recently published study in Taiwan. To clarify, it is important to distinguish the objectives and design of our study from that of the Taiwanese research. The Taiwanese study primarily aimed to investigate the association between concurrent PPI use and the prognoses (specifically, the risk of death) of CRC patients during cancer treatment. In contrast, our study focused on assessing the impact of previous PPI use history on the development of CRC in the Korean population. While both studies examine aspects of PPI use and CRC, they address different facets of the relationship between PPIs and death (Taiwanese study) or causes (our study) regarding to CRC. To address these difference, we added the following description to the revised Discussion section:
(Discussion, lines 368-385): To the best of our knowledge, in many countries where epidemiological investigations have been carried out, PPIs are available as over-the-counter drugs that can be obtained at pharmacies without the need for a doctor's prescription, with the exception of South Korea and Taiwan. The accessibility of and regulations surrounding the availability of PPIs can vary significantly between countries, and these differences may impact the patterns of PPI use and their potential effects. Given the variations in drug accessibility and healthcare practices across different countries, it is essential to consider how these factors might influence the relationship between PPI use and CRC risk. Our study in South Korea, where PPIs are prescription-based, provides a unique perspective within this specific healthcare context. In a recent study conducted in Taiwan, it was found that CRC patients who concurrently used PPIs with chemotherapy for more than 60 days faced a 1.10-fold elevated risk of cancer-specific death (95% CI, 1.01–1.17) [63], which underscores the potentially detrimental significance of PPI usage in CRC patients during cancer treatment. By examining the association between PPI use and CRC risk in a setting where these drugs are available exclusively by prescription, our study evaluating the impact of prior PPI usage history on the development of CRC in the Korean population may contribute to a more comprehensive understanding of the multifaceted interactions between PPIs and CRC.
Comment 5: line 366 - in my opinion additional limitation of the study is lack of data for patients, which have been used PPIs as OTC drugs, of course in the situation when it is possible in Korea. Therefore, in the Introduction it must be clearly written if such drugs are available as OTC, and a brief description in what regions of the world PPIs are available as OTC drugs.
Response: We appreciate the reviewer's valuable comment regarding the potential limitation related to the lack of data for patients who may have used PPIs as over-the-counter (OTC) drugs. While such usage is not permitted in Korea, which we have clarified in the revised Introduction, we believe that this is a valid limitation and have added it in the revised Discussion section:
(Discussion, lines 386-389) Given that this study solely concentrated on Korean citizens, extrapolating the findings to wider populations, especially those in regions where PPIs are available as over-the-counter medications obtainable at pharmacies without a prescription, may pose challenges.

Reviewer 2 Report
Comments and Suggestions for Authors
The authors have performed a thorough research of data base information regarding a possible association between PPI use and CRC in Korean population.
The first doubt is about methodology. The authors studied patients that have received PPIs within 365 days of developing CRC. From previous extensive research we know that CRC natural history requires many years to develop. The authors did not include patients that received PPIs before the one year period. I think this is a bias that needs further elaboration.
Are you suggesting that PPIs may produce a type of cancer with a different natural history than the one we have known up to now? Are you suggesting that there is some form of "acute" cancer development in contrast to the "slow" model we have used up to now?
If this is the case, I think further research is needed, particularly genotyping tumors.
The other point that creates doubts is that short treatments have a higher OR than long range treatments.
These doubts do not invalidate your findings, however, it suggests that there may be confounding factors that have not been adequately considered.
The topic you are treating in this paper is of high importance and if you have information about the genotype of these rapidly developing CRCs it would be of great significance to be included in your study.
Author Response
Reviewer #2:
General Comments: The authors have performed a thorough research of data base information regarding a possible association between PPI use and CRC in Korean population.
Response: We would like to express our sincere gratitude to the reviewer for dedicating their time and expertise to thoroughly evaluate our manuscript. Your constructive feedback is greatly appreciated, and we value your insightful comments.
Comment 1: The first doubt is about methodology. The authors studied patients that have received PPIs within 365 days of developing CRC. From previous extensive research we know that CRC natural history requires many years to develop. The authors did not include patients that received PPIs before the one year period. I think this is a bias that needs further elaboration.
Response: We sincerely appreciate your valuable comments and insights. You have pointed out the possible presence of a protopathic bias or the potential for reverse causality in our study. However, we wish to clarify that, in our research, we took measures to mitigate this bias by conducting adjustments for GERD episodes and H2-blocker use, as well as excluding diseases that could lead to gastrointestinal symptoms. We genuinely value your feedback and your suggestion to consider future research that includes patients with PPI use preceding the one-year period.
In response to your comment, we have incorporated the above explanation and limitation of our results in the revised Discussion and Limitation sections to provide transparency and context for our study:
(Discussion, lines 330-334): The studies alluded to the possibility of protopathic bias or the outcome of reverse causality, as the association between PPI use and CRC risk persisted for recent PPI use but not for past use. However, in this study, we found that both past and current PPI users had an increased likelihood of developing CRC, even after adjusting for GERD episodes and H2-blocker use and excluding diseases that cause gastrointestinal symptoms.
(Discussion, lines 390-395): While we adjusted for variables associated with PPI use to minimize confounding effects, the retrospective design and the reliance on diagnosis codes from Korean health insurance data, as well as the lack of information regarding CRC stage, location, phenotype, and genetic characteristics, may introduce limitations on our findings and potentially result in unmeasured confounding effects, thus making it difficult to apply our results to other demographic groups.
Comments 2 and 4: Are you suggesting that PPIs may produce a type of cancer with a different natural history than the one we have known up to now? Are you suggesting that there is some form of "acute" cancer development in contrast to the "slow" model we have used up to now? If this is the case, I think further research is needed, particularly genotyping tumors.
The topic you are treating in this paper is of high importance and if you have information about the genotype of these rapidly developing CRCs it would be of great significance to be included in your study.
Response: We greatly appreciate your thoughtful inquiry into the implications of our study's findings regarding the association between PPI use and incident CRC. Our study identified that both past and current PPI use were linked to a 1.19- and 6.06-fold higher likelihood of incident CRC when compared to the control group, even after making necessary adjustments. To provide a logical context for our results, we considered various theories that could potentially explain this association. While the majority of CRCs typically follow the adenoma-carcinoma sequence, which involves extended periods of development (e.g., 26 years for tubular adenoma, 9 years for tubulovillous adenoma, and 4 years for villous adenoma), there is an alternative theory of de novo colorectal carcinogenesis, characterized by rapid progression with a tumor doubling time of approximately 3 months. This theory has gained recognition and acceptance in numerous studies, particularly in the context of CRC development in Asia, where it accounts for a substantial portion (20% to 90%) of CRCs. Furthermore, we acknowledge the evidence from a mouse model, where mice administered PPIs for 4 weeks exhibited accelerated colon cancer growth. We refer to these theories and evidence to offer an evidence-based explanation for our results.
Regarding the suggestion to incorporate information about the location, phenotype, and genetic characteristics of CRCs, we regret to inform you that our national cohort database did not include such details. This limitation has been duly noted and discussed in the Limitations section of our manuscript:
(Discussion, lines 390-395): While we adjusted for variables associated with PPI use to minimize confounding effects, the retrospective design and the reliance on diagnosis codes from Korean health insurance data, as well as the lack of information regarding CRC stage, location, phenotype, and genetic characteristics, may introduce limitations on our findings and potentially result in unmeasured confounding effects, thus making it difficult to apply our results to other demographic groups.
Comment 3: The other point that creates doubts is that short treatments have a higher OR than long range treatments. These doubts do not invalidate your findings, however, it suggests that there may be confounding factors that have not been adequately considered.
Response: We sincerely appreciate your observation regarding the variation in odds ratios between short-term and long-term PPI treatments. While we have taken steps to adjust for potential confounding factors such as GERD episodes and H2-blocker use, and have excluded diseases that cause gastrointestinal symptoms, there may still be unidentified confounding variables at play. It is worth noting that in South Korea, PPIs can be prescribed not only for GERD but also for conditions like gastric ulcers or for critically ill patients hospitalized with acute illnesses. These scenarios may introduce confounding factors that could potentially increase the risk of CRC in short-term PPI treatments. However, as evidenced in our study, we believe that the consistent elevation in the risk of CRC, even after 90 to 360 days of PPI use, underscores the association between PPIs and CRC. We have described this limitation in the revised Discussion section:
(Discussion, lines 390-395): While we adjusted for variables associated with PPI use to minimize confounding effects, the retrospective design and the reliance on diagnosis codes from Korean health insurance data, as well as the lack of information regarding CRC stage, location, phenotype, and genetic characteristics, may introduce limitations on our findings and potentially result in unmeasured confounding effects, thus making it difficult to apply our results to other demographic groups.

Reviewer 3 Report
Comments and Suggestions for Authors
The Authors carried out a relevant study with a rigorous methodology in order to assess whether there is a relation between PPI use and the development of CRC in the Korean population. Their results seem to show that the use of PPIs is associated with an increased risk of CRC. I have several concerns about their conclusions as follows:
- The study has been performed in a very large sample size, but it has the important limitation of the retrospective design. The majoriy of studies on the presumed adverse events of PPIs are unfortunetely observational and retrospective and this is the reason of their questionable results (see Savarino et, Digestive and Liver Disease 2016). This limitation should be emphasized in the discussion of the paper.
- The increased risk of PPI use seems to be independent on the duration of this therapy. This is rather surprising because the odds of this risk decreases from short tretment (< 30 days) to the longer treatments. It is obvious that long-term therapy with PPIs maintains higher levels of gastrinemia and a constant modification of gut microbioma compared with short-term treatments. How can the authors explain this reduction of risk overtime ?
. it is correct that the authors have emphasized that their results are related to the Korean population because they cannot be generalized. Also this concept should be highlighted in the conclusions as the ethnicity might play a strong role in this effect.
Author Response
Reviewer #3:
General Comments: The Authors carried out a relevant study with a rigorous methodology in order to assess whether there is a relation between PPI use and the development of CRC in the Korean population. Their results seem to show that the use of PPIs is associated with an increased risk of CRC. I have several concerns about their conclusions as follows:
Response: We would like to express our sincere appreciation to the reviewer for their diligent evaluation of our manuscript. Your thoughtful feedback is of great value to us, and we are grateful for your constructive comments and concerns.
Comment 1&3: The study has been performed in a very large sample size, but it has the important limitation of the retrospective design. The majoriy of studies on the presumed adverse events of PPIs are unfortunetely observational and retrospective and this is the reason of their questionable results (see Savarino et, Digestive and Liver Disease 2016). This limitation should be emphasized in the discussion of the paper.
It is correct that the authors have emphasized that their results are related to the Korean population because they cannot be generalized. Also this concept should be highlighted in the conclusions as the ethnicity might play a strong role in this effect.
Response: Thank you for your insightful comments regarding our study. We acknowledge the limitations associated with the retrospective design, which is a common challenge in many studies examining the potential adverse events of PPIs. As you rightly pointed out, the majority of such studies are observational and retrospective in nature, which can introduce limitations in terms of establishing causality and drawing definitive conclusions. In our revised Discussion, we emphasize this limitation and acknowledge its potential impact on the interpretation of our results. We also highlight the importance of considering the specific Korean population in our conclusions and underscore the potential role of ethnicity in the observed effects:
(Discussion, lines 386-395): This study had some limitations. Given that this study solely concentrated on Korean citizens, extrapolating the findings to wider populations, especially those in regions where PPIs are available as over-the-counter medications obtainable at pharmacies without a prescription, may pose challenges. Ethnicity might also play a strong role in the effect of PPIs on CRC incidence. While we adjusted for variables associated with PPI use to minimize confounding effects, the retrospective design and the reliance on diagnosis codes from Korean health insurance data, as well as the lack of information regarding CRC stage, location, phenotype, and genetic characteristics, may introduce limitations on our findings and potentially result in unmeasured confounding effects, thus making it difficult to apply our results to other demographic groups.
Comment 2: The increased risk of PPI use seems to be independent on the duration of this therapy. This is rather surprising because the odds of this risk decreases from short tretment (< 30 days) to the longer treatments. It is obvious that long-term therapy with PPIs maintains higher levels of gastrinemia and a constant modification of gut microbioma compared with short-term treatments. How can the authors explain this reduction of risk overtime?
Response: While we performed adjustments for GERD episodes and H2-blocker use and excluded diseases that cause gastrointestinal symptoms to minimize bias, it is possible that there are other confounding factors at play. In South Korea, PPIs can be prescribed not only for GERD but also for conditions such as gastric ulcers or for critically ill patients hospitalized with acute illnesses. These different indications for PPI use might introduce variations in the risk of CRC. However, it is important to note that our study does show a consistent and elevated risk of CRC even after 90 to 360 days of PPI use, which suggests a continued relationship between PPIs and CRC risk. We have described this explanation in the Discussion and limitation section:
(Discussion, lines 330-335): The studies alluded to the possibility of protopathic bias or the outcome of reverse causality, as the association between PPI use and CRC risk persisted for recent PPI use but not for past use. However, in this study, we found that both past and current PPI users had an increased likelihood of developing CRC, even after adjusting for GERD episodes and H2-blocker use and excluding diseases that cause gastrointestinal symptoms.
(Discussion, lines 390-395): While we adjusted for variables associated with PPI use to minimize confounding effects, the retrospective design and the reliance on diagnosis codes from Korean health insurance data, as well as the lack of information regarding CRC stage, location, phenotype, and genetic characteristics, may introduce limitations on our findings and potentially result in unmeasured confounding effects, thus making it difficult to apply our results to other demographic groups.
